

# Ice Roughness Estimation via Remotely Piloted Aircraft and Photogrammetry

James Ehrman[1], Shawn Clark[1], and Alexander Wall[1]

[1]Department of Civil Engineering, 15 Gillson St., University of Manitoba, Winnipeg, MB R3T 5V6 Canada

**Correspondence:** James Ehrman (ehrmanj@myumanitoba.ca)

**Abstract.** Structure-from-Motion Photogrammetry conducted with images obtained via Remotely Piloted Aircraft (RPA) has revolutionized the field of land surface monitoring. RPA-Photogrammetry can quickly and easily capture a full 3D representation of a study area. The result of this process is a high-definition Digital Elevation Model (DEM) representing the land surface of a given study area. It is particularly useful in applications where land surface data collection would otherwise be expensive or dangerous. The monitoring of fluvial ice covers can be time-intensive, dangerous, and costly, if detailed data are required. Fluvial ice roughness is a sensitive parameter in hydraulic models and is incredibly difficult to measure directly using traditional field methods. This research hypothesized that the surface roughness of a newly-frozen fluvial ice cover is indicative of subsurface roughness. The hypothesis was tested through a comparison of ice roughness determined through the statistical analysis of RPA-photogrammetry DEMs to ice roughness values predicted by the Nezhikhovskiy equation. The Nezhikhovskiy equation is a widely used empirical method for estimating ice roughness based on observed ice thickness. Hydraulic and topographic data were collected over two years of field research on the Dauphin River in Manitoba, Canada. Various statistical metrics were used to represent the roughness of the DEMs. Strong trends were identified in the comparison of ice cover roughness values determined through RPA-photogrammetry and those calculated via the Nezhikhovskiy equation, as well as with ice thickness. The inter-quartile range of observed roughness heights was determined to be the most representative roughness metric. The maximum peak value performed better in some cases, but the fact that this metric would be heavily influenced by outliers led to it being rejected as a representative metric. Three distinct forms of surface ice roughness were noted: rough, smooth, and ridged. Statistical properties of the DEMs of fluvial ice covers were calculated. No DEMs were found to be normally distributed. k-means clustering analysis was used to group sampled data into two categories, which were interpreted as rough and smooth ice. The inter-quartile range of the smooth and rough categories were found to be 0.01 - 0.05 meters and 0.07 - 0.12 meters, respectively. RPA-photogrammetry was concluded to be a suitable method for the monitoring of fluvial ice covers. Other applications of RPA-photogrammetry for the characterization of fluvial ice covers are proposed.

## 1 Introduction

The consequences of ice on the flow regimes of rivers in cold climates can be dramatic, sometimes leading to loss of life and damage to infrastructure. In-stream infrastructure such as bridge piers, hydraulic control structures, and hydro-electric generating stations are subject to immense forces due to river ice; which is a critical factor in the design of such structures.



Understanding fluvial ice roughness is a critical step in better understanding the evolution and hydraulic impacts of fluvial ice covers. Currently, fluvial ice roughness is either estimated through empirical means, such as the Nezhikhovskiy (1964) equation, or through complex and expensive methods, such as hydraulic modelling. Direct measurements can also be made (Buffin-Belanger et al., 2015; Crance and Frothingham, 2008), or roughness can be inferred from a measured velocity profile 30 (Gerard and Andres, 1984). However, these direct measurement methods require personnel to conduct work on ice covers, which are frequently unsafe, and therefore limits the types of ice covers that can be studied.

The surface roughness of sea ice and land ice (typically glaciers) has been more extensively researched (Fitzpatrick et al., 2019; Dammann et al., 2018; Yitayew et al., 2018) than that of fluvial ice covers. This discrepancy is due in part to the scale of the these ice sheets, which allows for high-altitude remote sensing from manned aircraft using LiDAR and imagery and 35 satellites using synthetic-aperture radar (Dammann et al., 2018). The size and thickness of these ice formations also makes in-situ measurements generally more feasible from a safety perspective. The goal of obtaining roughness data for glaciers and sea ice surfaces often relates to the determination of aerodynamic roughness length, an important parameter in the estimation of heat fluxes (Fitzpatrick et al., 2019), although Dammann et al. (2018) evaluated sea ice roughness for the use of transportation planning.

An obvious solution to making the studying of fluvial ice covers safer is through the use of aerial vehicles. Helicopters, small fixed-wing aircraft, and satellites have long been used for the study of earth surface phenomena. All are prohibitively expensive to be solely dedicated to the study of ice covers, and none can produce images of sufficient resolution for surface roughness studies. Recently, Remotely Piloted Aircraft (RPA) have become much more accessible, inexpensive, and reliable. Coupled with high-resolution image-stabilized digital cameras, they offer the opportunity to document and study otherwise inaccessible 45 areas at a fraction of the cost of any other method. Structure-from-motion photogrammetry has been used extensively with RPA-acquired digital photos (RPA-Photogrammetry) (Colomina and Molina, 2014). The evaluation of surface roughness has also been studied using RPA-Photogrammetry on land surfaces (Kirby, 1991) and ice surfaces (Dammann et al., 2018; Chudley et al., 2019).

This research hypothesized that the surface roughness of a newly-frozen fluvial ice cover is indicative of subsurface rough-50 ness. The basis of this theory stems from field observations of ice mechanics on the Dauphin River, as observed by Wazney et al. (2018). Subsurface ice roughness investigations have been conducted on mature ice covers (Beltaos, 2013; Buffin-Belanger et al., 2015; Crance and Frothingham, 2008). It is likely that subsurface ice roughness measurements taken well after freeze-up will under-predict peak ice roughness due to flow smoothening of the subsurface over time. To the authors' knowledge, no investigations of subsurface ice roughness have been conducted on newly-frozen fluvial ice covers. The hypothesis was tested 55 while testing the capabilities of consumer-grade RPAs for winter river ice field work. The viability of photogrammetric analysis performed using a well-known professional photogrammetry software package, with images obtained using these RPAs was assessed. The statistical properties of Digital Elevation Models (DEM) obtained through these methods were examined and compared to roughness estimates obtained through the Nezhikhovskiy relationship. Finally, further applications of these tools to fluvial ice monitoring are proposed.





## 2 Background


The Dauphin River is located approximately 250 km North of the city of Winnipeg, in Manitoba, Canada, as shown in Figure 1. It drains lake St. Martin into Lake Winnipeg through 52 kilometres (km) of channel. The channel has steep, shallow banks that range between 110 - 160 meters (m) wide. The surficial geology of the area is composed of till with erratics, boulders, cobbles, and gravels observed throughout the channel. The most upstream 40 km of channel (Upper Dauphin River) has a mild

slope (0.029%) and is meandering. The bed composition of the Upper Dauphin River was observed to be primarily silt. The most downstream 12 km of channel (Lower Dauphin River) transitions into a well-defined riffle-pool system with a relatively higher slope (0.16%). Riffle sections in the Lower Dauphin River were observed to have a gravel-bed with some boulders and erratics. Pool sections were observed to be silt bottomed. During winter ice formation, dramatic ice consolidation events, jams, and flooding have been reported by Wazney et al. (2018) on the Lower Dauphin River. Lake Winnipeg water levels can have

a significant effect on the most downstream 2 km of this reach which is typically where the largest toe of the ice jam would form.

A Water Survey of Canada (WSC) gauge station (05LM006) is located ≈ 100 m downstream of site DRLL03, which logs water surface elevation and flow at five minute intervals, and reports daily values. The data are periodically adjusted for ice effects during the winter.

Roughness is an important parameter in the prediction of fluid flow along solid boundaries. This roughness creates drag along fluid boundary layers, generating the logarithmic fluid velocity distribution observed in open channel hydraulics. Nikuradse (1950) helped develop the concept of roughness height through equivalent sand grain roughness representing the roughness height of sand roughened pipes. More recently, an extensive discussion of methods used to represent the roughness of a heterogeneous three-dimensional surface layer from a surface profile was provided in Gadelmawla et al. (2002). Many of

these methods involve statistical analysis of the entire sample, or some subset (i.e. the peaks, valleys, etc.). Gomez (1993) used the difference between peaks and a locally-derived average bed surface for the investigation of gravel-bed roughness. Nikora et al. (1998) assumed their surface data derived from natural gravel point-bars constituted a random field, and found that the second-order moment of the frequency distribution yielded a suitable estimate of roughness height when compared to the Wolman (1954) method. The Wolman (1954) method is a widely-used field data collection method for the determination

of fluvial sediment grain-size distribution, typically in gravel-bed rivers. The method involves the semi-random, unbiased selection of fluvial bed material followed by the measurement of physical parameters of the material, typically b-axis length. Aberle and Nikora (2006) also investigated higher-order statistics, but confirmed the use of sample standard deviation (SD) as an appropriate representation of gravel-bed roughness height. For non-normal data, the inter-quartile range (IQR) is a more suitable representation of the spread of the data.

For hydraulic applications, conversion of roughness height values to Manning's n is desired. A widely-used quantitative method of estimating Manning's n from roughness height measurements was proposed by Strickler (1923). The Strickler coefficient ($c_n$) is the critical parameter of the Strickler equation. Strickler (1923) reported a value of $c_n = 0.047$ for general

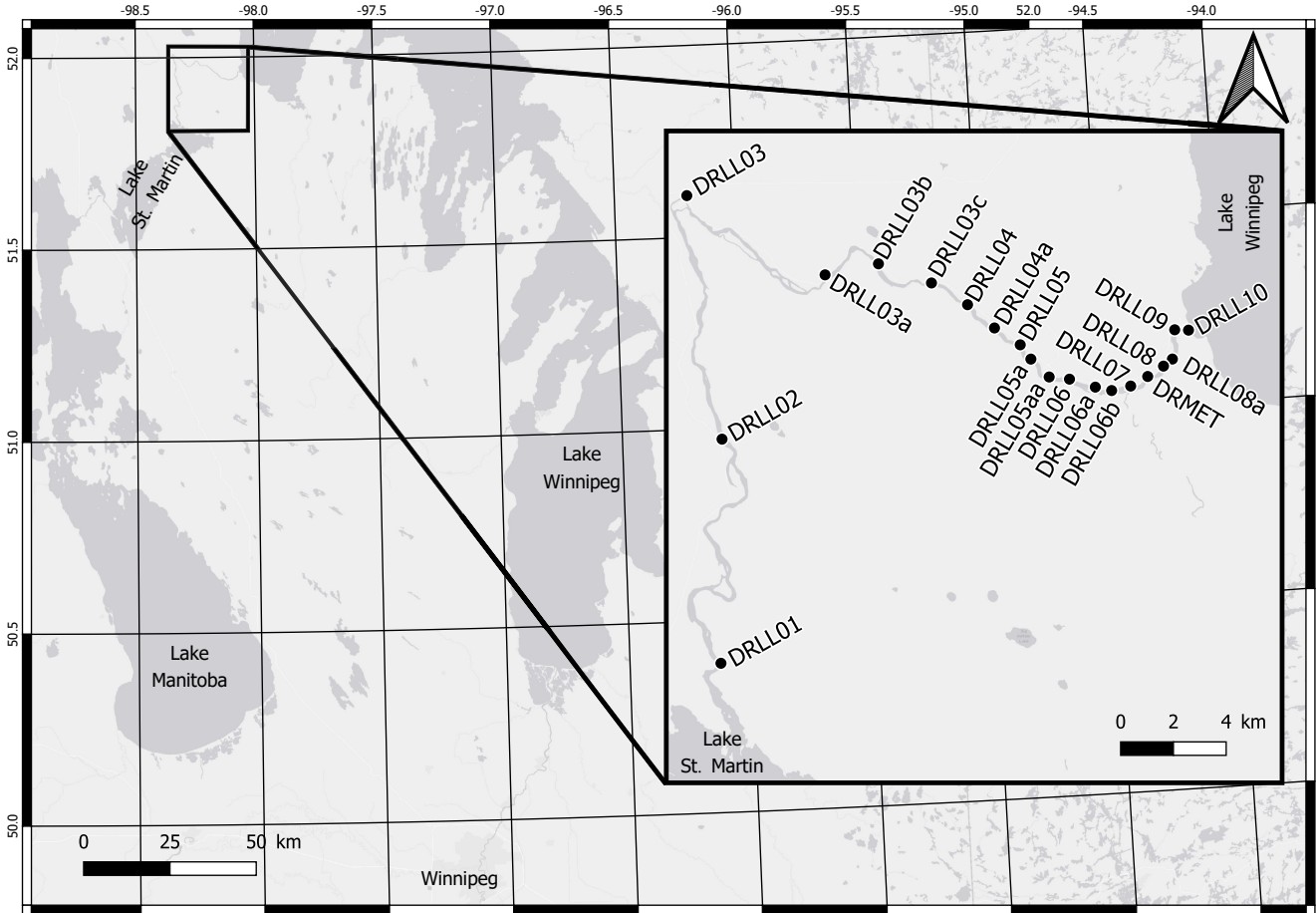

**Figure 1.** Key map of study location

use, but this can be adapted for specific applications (Sturm, 2001). The parameter $D[\mathrm{m}]$ represents the characteristic roughness height of the flow boundary.

$$n \approx c_n D^{1/6} \tag{1}$$

Fluvial ice formation has a significant impact on the roughness characteristics of Northern rivers. Perhaps the most consequential form of fluvial ice is frazil ice. Frazil ice may be formed once the water has become super-cooled, and the flow is turbulent and of relatively high velocity, above 0.5 - 0.9 m/s (Matousek, 1984). After generation, frazil ice will be transported downstream and either flocculate and rise to the surface to form pans of ice, or attach to the channel bottom forming anchor ice (Bisaillon and Bergeron, 2009). At this stage pans are very low density and mechanically weak (Beltaos, 2013). Frazil pans will then be transported further downstream. Depending on their length of travel and weather conditions, the portion of the pan





exposed to air may freeze and thicken adding strength. Finally, pans may jam against an obstacle such as an established ice cover.

Ice covers increase hydraulic resistance in fluvial systems by replacing the relatively friction-free air-water boundary with
a rougher ice-water boundary. This expands the wetted perimeter of the channel, and if the ice cover completely bridges the channel, may pressurize flow. The added source of roughness and constriction of flow results in upstream staging (Beltaos, 2013). As with estimates of channel boundary roughness, ice roughness can also be judged qualitatively based on general observations, with some success. The Nezhikhovskiy (1964) equation is widely used for this purpose, as illustrated in Equation 2, where $n_i$ is the Manning's roughness of the ice cover and $t_i$[m] is the cover thickness in m.

$$n_i \approx 0.0252 ln(t_i) + 0.0706 \qquad (2)$$

This relationship is based on measurements conducted on rivers in Russia several decades ago and it has served well as an estimation tool for engineering applications. Using more complex data, Equation 1 was adapted by Beltaos (2013) for use in the estimation of the roughness of newly-formed ice jams, resulting in Equation 3.

$$n_i \approx 0.095 D^{1/2} R^{1/3} \qquad (3)$$

The value given for $c_n = 0.095$ has been determined to be representative for ice jams. Additionally, the inclusion of the hydraulic radius $R$[m] accounts for the fact that the roughness elements of ice jams are often of such magnitude as to increase relative roughness to the point where it has significant impact on the hydraulic radius. This relationship is only valid for newly-formed ice jams. Immediately after formation, the ice is subject to shear forces from the water flowing underneath, which slowly smoothens the sub-surface of the ice cover (Ashton, 1986).

RPAs equipped with high-resolution digital cameras have been used extensively in the collection of near-surface photographic and topographic data (Colomina and Molina, 2014; Watts et al., 2012). They are smaller and more cost-effective than conventional aircraft allowing for much more versatile data collection. Compared to manual surveying methods they can collect a greater volume of data in less time and greatly reduce risk to personnel. For the purposes of topographic data collection, images collected with an RPA are subsequently processed using some form of photogrammetry. Niethammer et al. (2012) used
this method to monitor the progression of the Super-Sauze landslide, a task too dangerous to monitor manually. Eisenbeiss et al. (2005) employed RPA-photogrammetry to document the layout of ancient ruins in Peru. Completing this task manually would have risked the integrity of the site. Hamshaw et al. (2019) found use for RPAs in the monitoring of river-bank erosion. RPAs were even used by Alfredsen et al. (2018) in the mapping of river ice in Norway.

There are several methods of producing a DEM from a set of overlapping close-range aerial images. Currently, the most
widely used and time-effective method is that of Structure from Motion (SfM) photogrammetry (Matthews, 2008; Fraser and Cronk, 2009). For the purposes of this research, SfM photogrammetry was the sole method employed.

An important consideration when using the RPA-photogrammetry method for topographic data collection is the impact of doming errors on the final product (James and Robson, 2014). Doming errors are most prominent when all images are taken





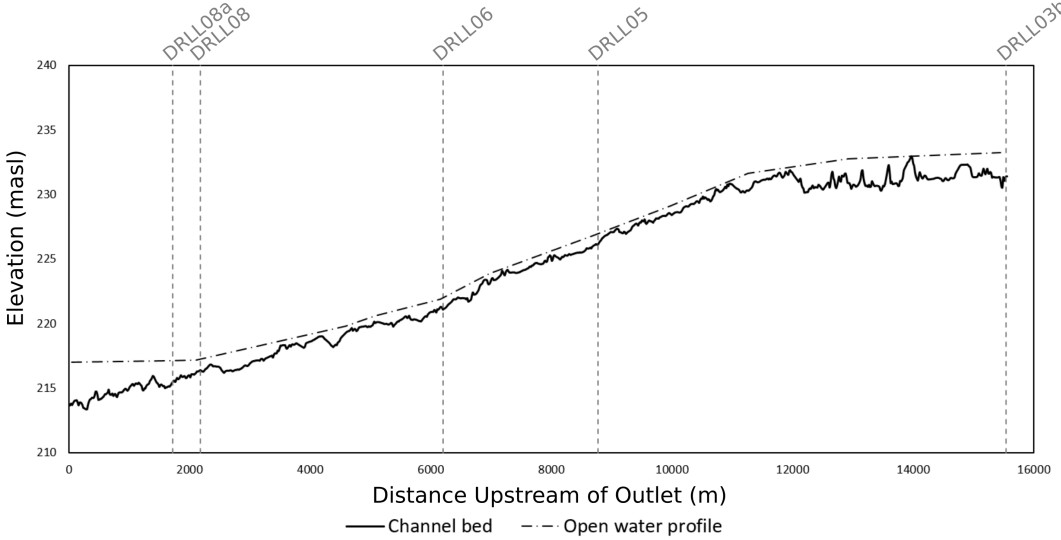

**Figure 2.** Channel bed profile of the Lower Dauphin River, with selected study locations, flow 86 m$^3$/s

from a parallel axis (Eltner and Schneider, 2015). In the case of RPA-photogrammetry, this is when the camera angle is set
to 0° tilt, however, some distortion is also caused by the shape of the camera lens. Most advanced software packages used to
produce DEMs from photogrammetry data include a self-calibration process that develops a model of the distortion caused by
the lens of the camera. Eltner et al. (2016) makes the distinction between local surface quality and more systematic errors such
as doming, relating these two categories to the precision and accuracy of the DEM, respectively.

## 3    Methodology

Five field sites were selected in this study, their relative location along the bed profile of the Lower Dauphin River is illustrated
in Figure 2. Data were gathered during the winter months of 2017 - 2019. A relatively smooth, unconsolidated ice cover has
been observed to form at DRLL03b in all previous study years, due to it's low bed slope (0.029%). Sites DRLL05 and DRLL06
exhibited substantial ice dynamics, as they are within the higher gradient (0.16%) Lower Dauphin River (Wazney et al., 2019).
Sites DRLL08 and DRLL08a had much milder water surface slopes, due to the backwater effect from Lake Winnipeg. The toe
of an ice jam has formed in previous years near sites DRLL08a and DRLL08. These sites were selected in an effort to compare
the efficacy of the RPA-photogrammetry method on different ice conditions, and to determine if the methods can distinguish
roughness differences between sites.

### 3.1    Field Methods



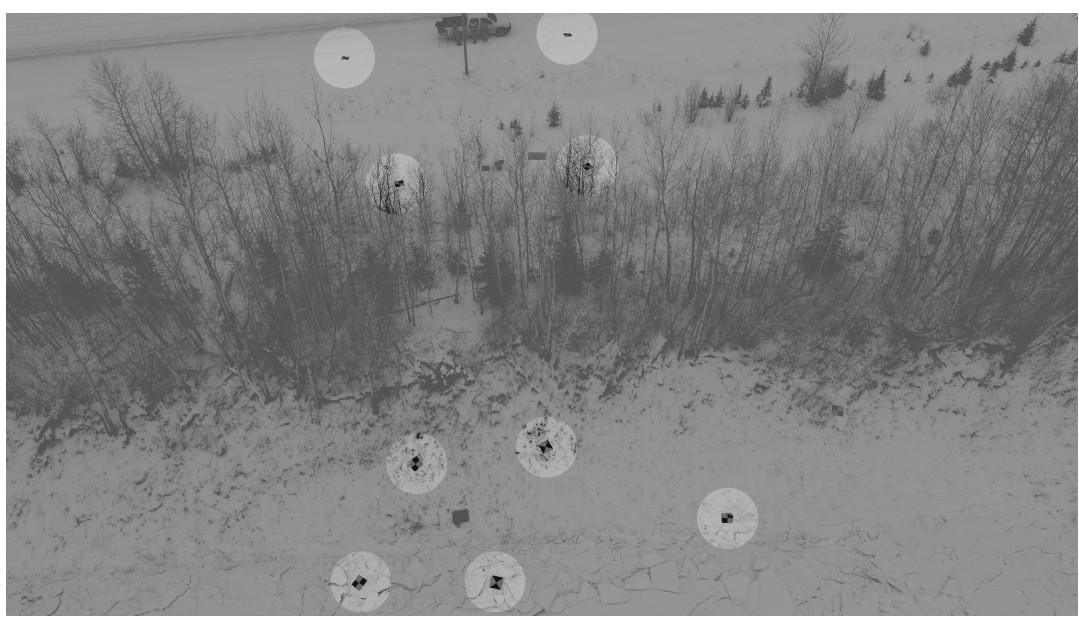

**Figure 3.** Typical target distribution

### 3.1.1 Photogrammetry

Ten 1 m$^2$ high-visibility medium-density fiberboard targets were distributed on the grounded ice near the left bank, and on snow near Provincial Road (PR) 513. A typical layout of targets is shown in Figure 3, which illustrates how the targets are placed exclusively on the left bank of the river. The targets were clustered in this way since the right bank was inaccessible. Ideally, the targets would have been evenly distributed across the entire study area (Alfredsen et al., 2018; Gini et al., 2013). In Section 3.1.3, the effects of target distribution on DEM accuracy are tested.

After targets were placed, their centres were surveyed using a Leica Viva GS14® survey-grade Real-Time Kinematic (RTK) Global Navigation Satellite System (GNSS) base-and-rover system, which is typically observed to have an in-field reported horizontal error of ≈ 2 cm, and a vertical error of ≈ 3 cm. The Canadian Geodetic Vertical Datum of 2013 (CGVD2013) geoid was used in the recording of all surveyed elevations. Localization was assessed using a Manitoba Infrastructure (MI) benchmark located near DRLL03, and verified using the Natural Resources of Canada Canadian Spatial Reference System

Precise Point Positioning (CSRS-PPP) service. Further benchmarks were established using the CSRS-PPP and "leap-frogging" to further benchmarks. In the 2019-2020 season, some RPA flights were completed without targets, to allow for more flights to be completed during the field visit. A comparison between the representative metrics calculated from a DEM with and without geo-rectification is presented in Section 4.1.

Once all field personnel had finished active tasks, the RPA was launched, and field staff remained stationary for the duration

of the RPA flight. A DJI Phantom 4 Professional® RPA was flown at an approximate altitude of 30 m, with overlapping photos taken every 10 m, at a 0° or 20° camera tilt. The on-board 20 mega pixel camera had an 84° field of view with a 1 inch CMOS





sensor. The RPA flight transected the river and included PR 513 and forest on the left and right bank. The RPA was flown only if wind speeds measured by a hand-held digital anemometer were less than 36 kilometres per hour (km/h). Further, light conditions could drastically impact the quality of images taken; the RPA was flown only during daytime and during clear, or
lightly overcast conditions. Typical capture dimensions of an RPA flight were 90 m in the stream wise ordinate, and 230 m across the river. Ideally, geo-rectification targets used in RPA-photogrammetry would be evenly distributed across the study area. In this research, targets could only be placed on the left bank of the study area, and none could be placed directly on floating ice, due to safety concerns.

During the 2019-2020 field season, the RPA mission planning application Pix4Dcapture® was used to plan and automate
RPA flights over study areas. This greatly reduced the required flight time, and produced similar, if not better photo coverage.

### 3.1.2 Hydraulic Parameters

Water pressure was recorded every eight minutes at the study sites using Solinst Levelogger® Edge 3001 M5 pressure transducers, and accompanying nearby Solinst Barologger ® Edge 3001. The listed accuracy of these devices is ± 0.003 m and ± 0.05 kPa respectively. These instruments were installed before the ice season began (typically October), removed for download and
maintenance after the end of the ice season (typically April/May), and were then subsequently re-installed for summer observations. During installation, the water surface was surveyed for use in post processing to determine the absolute water surface elevation of the observations in meters above sea level (masl). Additionally, the observed barometric pressure, converted to its equivalent depth of water, was subtracted from the pressure observations. The observed water level and previously measured channel bathymetry were used to estimate hydraulic radius.

Late in the winter (typically February) after a stable ice cover had formed, an ice elevation survey was undertaken. Using the base-and-rover system, the top of ice elevation was surveyed along the length of the Lower Dauphin River. Truncated transects of ice thickness were also surveyed at locations where ground elevation had previously been surveyed. A transect was performed at site DRLL06 and DRLL05, but not at DRLL08, due to safety concerns. During the 2019-2020 field season holes were drilled in the ice cover at safe locations, and following established safe work procedures, to determine ice thickness. If
measurements of ice thickness were not possible at a given site, for a given year, thickness was estimated through observation and photographs.

### 3.1.3 Field Accuracy Tests

There was a need to quantify the impact of the ground control targets being clustered on the left bank of the study area. The field methods described in Section 3.1.1 were repeated at River's Edge Nursery in La Barriere, Manitoba. A fully dry land study area
of equivalent size to typical study areas flown at the Dauphin River was delineated, and 15 targets were distributed. The targets were conceptually grouped into three areas: typical, middle, and end. The "typical" group represented a target distribution that was generally produced during field work at the Dauphin River sites. The "middle" and "end" target groups were supplemental, which would be added or subtracted from the photogrammetry analysis to test their respective impacts on DEM accuracy. The





distribution of targets in the study area is represented in Figure 4. Finally, after the RPA flight was conducted, 10 independent
and unmarked locations were captured by RTK-GNSS survey as a check for accuracy in subsequent data analysis.

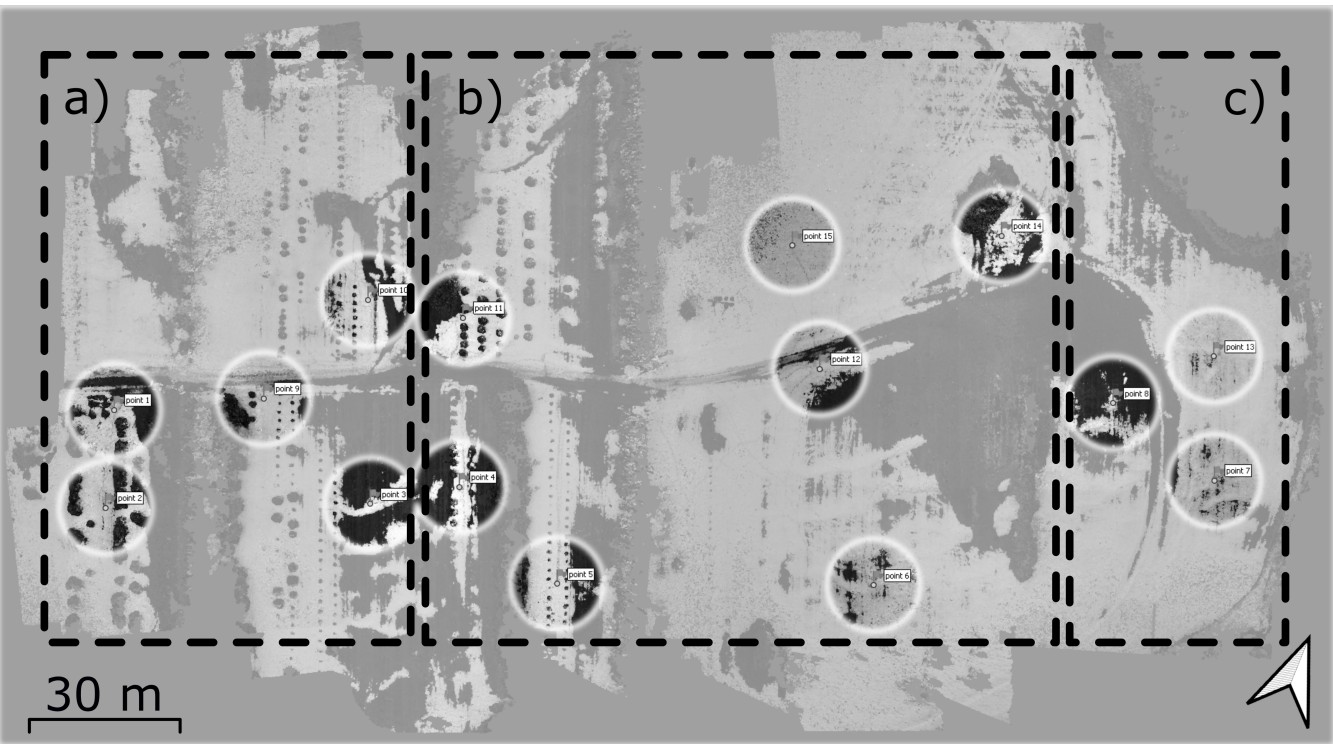

**Figure 4.** Accuracy test experimental set up, a) typical, b) middle, c) end

## 3.2 Laboratory Methods

### 3.2.1 Photogrammetry

PhotoScan Professional® from AgiSoft LLC uses the SfM method of photogrammetry, which has been widely used in geo-
sciences in recent years (Westoby et al., 2012), as well at in this study. Gini et al. (2013) compared their custom research-
grade photogrammetry algorithms to results obtained from Pix4UAV Desktop® and PhotoScan Professional®. Their findings
suggested that these commercial packages performed similarly to their software, with PhotoScan Professional® performing
somewhat better than Pix4UAV Desktop®. PhotoScan Professional® is also considered to be a relatively fully-featured and
complex (Eltner and Schneider, 2015) tool as compared to other options, and includes an automated process for estimating and
correcting doming errors.

Images were imported into PhotoScan Professional®, and aligned to create a sparse point cloud of tie points. Where targets
were used, they were identified in all images containing them, and their coordinates were imported to provide geo-rectification



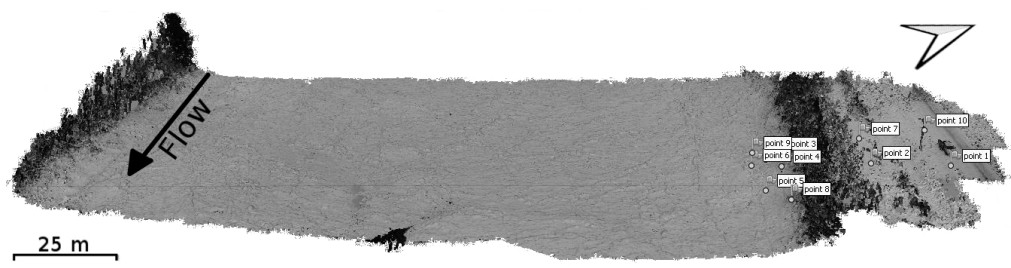

**Figure 5.** Example point cloud, DRLL06 2018-11-21

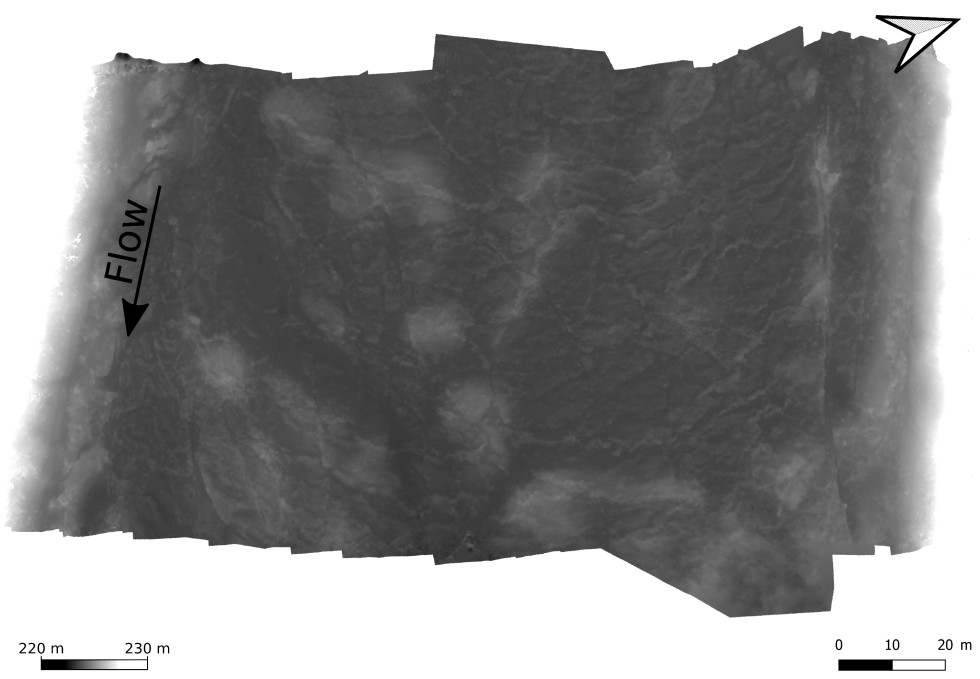

**Figure 6.** Example DEM, DRLL06 2018-11-21

of the resultant point cloud. A dense point cloud was then generated, followed by a DEM. An example point cloud consisting of $\approx 15$ million points and corresponding DEM are shown in Figures 5 and 6.

### 3.2.2 Accuracy Testing

The impact of placing all control points on only one bank was tested through a detailed trial on an open field. Groups of targets were used as input to the photogrammetry software and the resultant DEMs were compared to the 10 independent survey points. The DEM generated using all available targets was assumed to be the most correct representation of the land surface, against





which all other target groupings were compared. A maximum acceptable vertical difference of 0.03 m between the DEMs and the independent survey points was adopted. This value was chosen based on the typical error observed in the data gathered by the RTK GNSS base-and-rover system. This system was the limiting factor for accuracy in this study since it was the tool which informs the absolute spatial position of all field equipment. The following target scenarios were tested: "all points" utilizing all ground control targets, "typical points" using all the targets identified in the "typical" subset, "three points" using a subset of three targets from the "typical" subset, and "two points" using a subset of two targets from the "typical" subset. In the two and three point tests, the most spatially distributed targets within the "typical" subset were selected. An additional test was required to determine if systemic errors were introduced in DEMs generated without the use of geo-rectification targets. The data collected at site DRLL06 on 2019-11-13 was prepared with and without the inclusion of control point data. A maximum acceptable percent error of 5% was adopted to evaluate the results of this test.

### 3.2.3 Roughness Characterization

To avoid unwanted influences in the surface slope and texture, a 50 m$^2$ sub-sample from the center of the river was taken, which excluded all overbank objects and sections of the ice cover that were near to the bank. Additionally, a three-dimensional plane-of-best fit Linear Model (LM) was found for each sample, and then subtracted from the surface data. The goal of this was to normalize each data set, setting the average surface elevation to 0, and removing the river slope from the sample data. Gadelmawla et al. (2002) noted that the average surface elevation is the most commonly used, and most sensible reference standard from which to assess roughness height. By shifting the elevation data down to a base elevation of 0, and removing unwanted patterns, each data point was transformed from an elevation to a roughness height.

A two-dimensional Fast Fourier Transform (FFT) was then applied to each sub-sample, with the goal of filtering the input data and removing other surface trends beyond those addressed with the plane-of-best-fit. The combination of the LM and FFT adjustment and filtering process will be referred to as LMFFT. Though an analysis of dominant frequencies it was found that the lowest frequencies ($< 1$ m$^{-1}$) had the largest amplitudes, while the highest frequency signals ($> 5$ m$^{-1}$) had the lowest amplitudes. A typical distribution of amplitudes is shown in Figure 7 a. These results are arranged such that the highest frequency waves are found in the center of the matrix, while the lowest frequencies are found in the corners of the matrix. Based on this interpretation, a band-pass filter was constructed as an annulus about the center of the results matrix, the result of the application of this filter is shown in Figure 7 b. Each point in the results matrix was interpreted to have a frequency based on its distance from the center using the spatial coordinates of the original image. A low-pass filter value of 0.08 m was generally found to produce the best results. The high-pass component of the filter was adjusted for each image, to ensure large trends were removed. The wavelength cut-off ranged from 70 m to 70.5 m. Figure 7 c and d show the DEM before and after the application of this filter.

Once the data were in a form that directly represented roughness height, and unwanted patterns had been removed, assessment of statistics representing bulk roughness were computed. Based on a review of relevant literature in the fields of photogrammetry, fluvial geomorphology, and roughness characterization, various statistical methods were considered, and several were chosen for further consideration in this study (Table 1). The data were grouped into three categories: the first being the





**Figure 7.** Summary of impacts of filtering through Fourier analysis, DRLL06 2019-11-13 a)Fourier amplitude plot, b) Filtered Fourier amplitude plot, c) Original ice surface DEM, d) Filtered ice surface DEM

"Raw" data, which was the set of DEM data as reported by the RPA-photogrammetry technique; the second is referred to as the "General" data, which was all the DEM data after linear and FFT analyses have been conducted; and the "Peaks" data, which corresponded to values of peaks within the sample. Peaks were determined by evaluating all one-dimensional cross section of



**Table 1.** Selected statistical methods

| Method | Data Group | Reference |
|---|---|---|
| IQR | General | Aberle and Nikora (2006) |
| SD | General | Aberle and Nikora (2006) |
| Minimum Peak Value | Peaks | Gadelmawla et al. (2002) |
| Maximum Peak Value | Peaks | Gadelmawla et al. (2002) |
| Average of Peaks | Peaks | Gadelmawla et al. (2002), Gomez (1993) |
| 84th Percentile of Peaks | Peaks | Beltaos (2013) |

the data (rows and columns) of the data, and applying a one-dimensional peak picking algorithm. The identified peaks were filtered, selecting only those elements which were found to contain a peak in both it's associated row and column.

The roughness heights found from the above methods were then used to calculate Mannings n for each value, using Equation 3 (Beltaos, 2013), which is an adaptation of the Strickler formulation, and has been used for ice jam roughness. Additionally the hydraulic radius, adjusted for ice thickness was found using the observed water level, ice thickness, channel cross section,
and assumed specific gravity of ice of 0.916.

In addition to numerical characterization of roughness, classification of roughness type was undertaken using k-means clustering, which is the most commonly-used clustering approach (Jain, 2010). The data used for input were the following: IQR of the general and peak data, median of the peak data, and kurtosis of the general and peak data. Mean and median of the general data were excluded since they were set to 0 through the linear adjustment and Fourier analysis. The mean of the
peak data was excluded since it is the first statistical moment, and is related to the kurtosis. Since it is used in the computation of kurtosis, SD and skewness was removed from both data sets. The data were then mean-centred and the optimal number of clusters was determined using the average silhouette method. The Euclidean distance formula was used in determination of the k-means clustering.

### 3.2.4 Roughness Comparison

Roughness metrics were compared with the following scheme with two schemes. The first being Nezhikhovskiy ice Manning's n versus observed ice Manning's n (Nez. - n). The second being Ice Thickness vs. observed ice Manning's n (Thick. - n). The ice Manning's n predicted through the Nezhikhovskiy relationship was compared to observed ice Manning's n to provide context for the relationship between observed roughness and a widely-used method of determining ice roughness. Since ice thickness is the sole input parameter of the Nezhikhovskiy relationship, it was compared to observed ice Manning's n to determine if there
may be any relationship between surface ice roughness and ice thickness. A relationship between these two parameters would indicate a relationship between surface and subsurface ice roughness. All comparisons in each item of the above scheme were made against the six roughness height metrics defined in Section 3.2.3. This resulted in two sets of six plots. A linear regression was computed for the data in each plot. The quality of regressions were evaluated using the $R^2$, $p$, and root-mean square error



**Table 2.** Percent and absolute difference between various statistical metrics computed from geo-rectified and non geo-rectified DEMs produced from data collected and site DRLL06 on 2019-11-13

|          | Max.   | Min.   | Mean     | Median   | IQR      | SD       | Skew. | Kurt. |
|----------|--------|--------|----------|----------|----------|----------|-------|-------|
| Absolute | 0.02 m | 0.12 m | <0.01 m  | <0.01 m  | <0.01 m  | <0.01 m  | 0.02  | 0.08  |
| Percent  | 4      | 43     | <1       | 6        | 2        | 2        | 5     | 2     |

(RMSE) values. The criteria for significance was set to $\alpha \leq 0.05$. In the case of the Nez. - n comparison a one-to-one line was

also plotted on the chart, and RMSE was computed between the observed ice roughness values and their associated modelled

value. The goal of the RMSE value was to evaluate the error between the six plotted metrics in each scheme. The validity of

using linear regression modelling with these data was evaluated by testing the normality of residuals using the Shapiro-Wilk

test for normality ($p > 0.05$). This test was selected due to it's wide usage in data analysis and it's superior power to many

other widely-used normality tests (Razali and Wah, 2011). Q-Q plots of residuals were used to confirmed the test statistics.

## 4 Results

### 4.1 RPA Performance

The RPA chosen for this study, the DJI Phantom 4 Professional[®], performed well during all field visits in various weather

and cloud conditions. In extreme cold (-20°C and below) the RPA performed all functions well, although the battery life was

reduced by approximately 50%. Additionally, it was found that the RPA had to be powered on in a warm area, such as the

heated cab of the field vehicle. Once powered on, it could then be placed outside for normal operations.

RPA-Photogrammetry performed very well across a variety of scenarios in the land-based field accuracy tests. The worst

observed accuracy was found in the "two points" scenario with an average vertical difference of 6.30 m calculated between

the DEM and the 10 independently surveyed test points. The "three points", "typical points", and "all points" scenarios all had

the same average vertical difference of 0.03 m. These three scenarios were all within the acceptable vertical difference criteria

of 0.03 m identified in Section 3.2.2. The maximum error of the DEM was determined to be the sum of the expected vertical

accuracy of the RTK-GNSS unit, 0.03 m, and the observed vertical difference difference between the DEMs and surveyed

locations, 0.03 m, for a total expected error of 0.06 m. The results of the test to determine if a lack of geo-rectification targets

introduces systemic errors into the DEM data are presented in Table 2.

### 4.2 Dauphin River Results

During the 2018-2019 and 2019-2020 field seasons, the Dauphin River experienced much lower flows than those observed

in the previous few years. The mean seasonal flow between November and March for each season was 74 and 90 m³/s for

2018-2019 and 2019-2020 respectively, compared to 178 and 195 m³/s in 2016-2017 and 2017-2018 respectively, although in





**Table 3.** Statistical properties of general DEM data obtained through RPA-photogrammetry

| Site | Date | Mean (m) | Median (m) | IQR (m) | SD (m) | Skew. (-) | Kurt. (-) |
|---|---|---|---|---|---|---|---|
| DRLL03b | 11/14/2019 | <0.01 | <0.01 | 0.01 | 0.02 | 2.04 | 15.51 |
| DRLL05 | 11/12/2019 | <0.01 | <0.01 | 0.02 | 0.02 | 0.79 | 6.45 |
| DRLL05 | 11/13/2019 | <0.01 | <0.01 | 0.02 | 0.02 | 0.73 | 7.59 |
| DRLL06 | 11/21/2018 | <0.01 | -0.01 | 0.12 | 0.10 | 0.48 | 3.91 |
| DRLL06 | 2/20/2019 | <0.01 | -0.01 | 0.12 | 0.11 | 0.96 | 5.81 |
| DRLL06 | 11/12/2019 | <0.01 | <0.01 | 0.08 | 0.06 | 0.32 | 4.32 |
| DRLL06 | 11/13/2019 | <0.01 | <0.01 | 0.08 | 0.07 | 0.44 | 3.68 |
| DRLL06 | 11/26/2019 | <0.01 | <0.01 | 0.07 | 0.06 | 0.46 | 3.78 |
| DRLL08 | 11/13/2019 | <0.01 | <0.01 | 0.05 | 0.04 | 0.88 | 5.70 |
| DRLL08a | 11/13/2019 | <0.01 | <0.01 | 0.05 | 0.05 | 0.98 | 6.64 |

prior years lower flows were noted. This resulted in notably thinner ice covers, more thermal ice growth, and less extensive ice jamming than was reported by Wazney et al. (2018). The average values of observed ice thickness reduced from 2.9 m at site

DRLL05 and DRLL06 in 2017-2018 to 1.8 m and 0.8 m in 2018-2019 and 2019-2020 respectively.

### 4.2.1 Statistical Properties of Ice Roughness Height Distributions

Several different forms of ice roughness were observed in DEMs produced using RPA-photogrammetry. Figure 8 illustrates three different ice roughness forms and their appearance in cross section along the indicated transects. The "rough" form of ice roughness was classified visually as any type of ice formed by the accretion and consolidation of frazil pans, but without

extensive secondary formations, such as coherent compression ridges. The "smooth" form of ice roughness was classified as ice that appeared to have formed under quiescent conditions, from a combination of transported and thermally-grown ice that did not consolidate. Ice which exhibited pressure ridges on otherwise smooth ice was termed "ridged".

Ridged ice presented a unique situation for the evaluation of ice roughness based on surface ice characteristics. In the fluvial setting the relationship between the height of an observed pressure ridge above the ice cover (sail), and the depth to which

the ridge extends below the ice cover (keel). In sea ice, the keel is greater than the sail (Johnston et al., 2009). Two samples were observed to display ridged ice, both of which occurred at site DRLL08a, on the dates 11/23/2017 and 11/26/2019. These samples were discarded from subsequent analysis.

Tables 3 and 4 outline the results of analyses of the roughness height distributions for the raw data, the general transformed DEM data, and the peak data (a subset of the general data), respectively.

Cluster analysis was conducted using k-means clustering. The optimal number of clusters was found to be two using average silhouette analysis. This was enforced by observations taken at the time of sample collection. The samples were broadly separated into two groups corresponding to the visual extent of "rough" or "smooth" ice, as defined earlier in this section. All



**Table 4.** Statistical properties of peak data obtained through RPA-photogrammetry

| Site | Date | Mean (m) | Median (m) | IQR (m) | SD (m) | Skew. (-) | Kurt. (-) |
|------|------|----------|------------|---------|--------|-----------|-----------|
| DRLL03b | 11/14/2019 | 0.04 | 0.03 | 0.02 | 0.02 | 2.63 | 12.86 |
| DRLL05 | 11/12/2019 | 0.04 | 0.03 | 0.01 | 0.01 | 2.13 | 9.64 |
| DRLL05 | 11/13/2019 | 0.04 | 0.04 | 0.02 | 0.02 | 2.08 | 10.29 |
| DRLL06 | 11/21/2018 | 0.25 | 0.23 | 0.08 | 0.06 | 1.46 | 5.67 |
| DRLL06 | 2/20/2019 | 0.25 | 0.22 | 0.07 | 0.07 | 2.40 | 10.62 |
| DRLL06 | 11/12/2019 | 0.16 | 0.15 | 0.05 | 0.05 | 2.34 | 11.24 |
| DRLL06 | 11/13/2019 | 0.18 | 0.16 | 0.05 | 0.05 | 1.55 | 5.71 |
| DRLL06 | 11/26/2019 | 0.15 | 0.14 | 0.04 | 0.04 | 1.99 | 9.38 |
| DRLL08 | 11/13/2019 | 0.10 | 0.09 | 0.03 | 0.03 | 2.46 | 14.54 |
| DRLL08a | 11/13/2019 | 0.12 | 0.10 | 0.06 | 0.05 | 1.69 | 6.29 |

observations at site DRLL06 were found to belong to cluster "1", while observations at site DRLL03b, DRLL05, DRLL08, and DRLL08a belonged to cluster "2". The within cluster sum of squares was found to be 5.44 and 11.96 for clusters 1 and 2

respectively.

## 5 Discussion

### 5.1 Accuracy of the RPA-photogrammetry Method

Generally, if three or more targets were used, vertical differences of no more than 0.03 m were observed between resultant DEMs and independently surveyed points. This amount of vertical difference was deemed acceptable when compared to the

maximum acceptable vertical difference of 0.03 m established in Section 3.2.2. No excessive tilt was observed in DEMs as a result of clustering targets on one side of the study area. AgiSoft PhotoScan Professional® appeared to be able to find an adequate number of tie point between images principally comprised of snow. The photogrammetry algorithm generally struggled to resolve points within forested areas. This limitation is well known within photogrammetry (Harwin and Lucieer, 2012; Hamshaw et al., 2019; Lane et al., 2000), but since there were no trees in the areas of interest of our study areas, this

limitation was not consequential to the study.

As described in Eltner et al. (2016) systemic errors causing deficiencies in DEM accuracy differ from local-scale errors causing deficiencies in DEM precision. Systemic errors include those incurred by improper sampling technique and by limitations in the analysis. These errors were largely assumed to have been handled to the extent that is possible in this research by the automated processes in AgiSoft PhotoScan Professional®. Eltner and Schneider (2015) found that AgiSoft PhotoScan

Professional® also performed well in reproducing the texture of complex natural surfaces. Direct comparisons could not be made in this research between the naturally occurring ice surfaces and the RPA-photogrammetry reproductions. However, the





magnitude of such results as the maximum ice peak value matched visual observations and field notes. The accuracy test performed at the La Barrier field site confirmed that with appropriate ground control points this method could accurately reproduce snow-covered land surfaces. It also showed that the method could precisely reproduce features of the same order of magnitude as the chunks of ice expected to be measured on the Dauphin River.

The additional test performed to determine if systemic errors were introduced in DEMs lacking geo-rectification resulted with most metrics well within the 5% threshold. The median had percent differences greater than 5%; however, it's absolute deviation was $< 0.005$ m, which is much less than the maximum vertical error (0.06 m). The minimum value deviated by 43% and 0.12 m. This deviation puts the precision of the extreme values into question. However, since the mean, median, SD, and IQR all had deviations $< 0.005$ m, it was concluded that DEMs which were generated without geo-rectification targets were not subject to systematic errors which could greatly impact the precision of the overall data set.

### 5.2 Statistical Properties of Ice Surface Roughness Heights

The distribution of roughness heights changed substantially through the stages of analysis. Initially, the distributions were somewhat non-normal, generally with peaks slightly to the left of the mean value (0 m), or bi-modal, such as in some cases at site DRLL08a. After the FFT had been conducted, the data appeared to be more normal, with slightly less biased peaks, and more even tails. The data were tested for normality using the Shapiro-Wilk test. All tested distributions failed the Shapiro-Wilk normality test, with $p << 0.05$ (n $\approx$ 5000). Considering the distribution of peak values only, all sites exhibited clearly non-normal distributions, heavily biased to the extreme left of the chart. This was interpreted to mean that the majority of peaks are small compared to the maximum peak values. It also means that the mean of peaks value is probably not an accurate physical representation of the data set.

In some cases where RPA flights had been conducted at the same sites on consecutive days, differences were observed between the distributions of roughness heights and peaks. The distribution of raw and general data at sites DRLL05 changed between 2019-11-12 and 2019-11-13 (referred to as the 12th and 13th for the remainder of this paragraph). The distribution of raw data became bimodal and the distribution of general data had longer tails on the 13th. When the representative photos included in the charts were compared, it could be seen that by the 13th, the ice cover was more opaque. Water level data showed that a small consolidation event likely occurred between the two tests, presumably changing the nature of the surface ice. The peak data remained somewhat consistent, although more peaks were noted on the 13th. At site DRLL06, the tails of the distribution were longer on the 12th than the 13th, although the overall distribution shape and height were fairly consistent. Some of these differences could be explained by differences in light conditions between the RPA flights, indeed this may have given rise to some of the differences observed at site DRLL05 over the same time period. This was corroborated by the RPA flight conducted at DRLL06 on 2019-11-26, under similar light conditions to the flight on the 13th, which showed very similar general and peak distributions. Similar general and peak distribution data were noted at site DRLL08a between the two RPA flights conducted on the 13th and 2019-11-26. The raw distributions had some noted differences in composition, although they were similar in height and width.



Using the results of the k-means clustering analysis, a breakdown of ice roughness cluster characteristics was generated. Table 5 illustrates the site grouping under the cluster analysis results, with maximum, mean, and minimum values reported within each cluster. The two best performing roughness metrics in the above subsections, IQR and 84th percentile of peaks are included, as well as the strongest defining metrics in the principal components analysis, IQR and kurtosis are included in this table. This table may serve as a defining guide for further categorization of ice surface types. Since cluster "1" corresponded

to ice surfaces with higher IQR and 84th percentile of peaks values, this group was interpreted as the "rough" ice group. Consequently, cluster "2" was termed "smooth". The range of values are mostly mutually exclusive, indicating a strong divide between the two groups.

Histograms of roughness height distributions were generated for each sample site. Using the groups identified by the k-means clustering analysis, and including a new group of ridged ice samples, 3 representative samples were selected, their

roughness height distributions are illustrated in Figure 9. A representative image of the ice DEM as well as the distribution of peak roughness height values is also included. It was noted that the rough and smooth samples differed primarily in the width and height of their distributions. The rough samples tended to have wider distributions with a lower peak count, while the smooth distributions had a higher peak count and a narrower distribution of roughness heights. The distributions of peak roughness height also differed, with the rough samples having more larger peaks than the smooth samples. The ridged samples

exhibited more irregular distributions than the rough or smooth cases, but were more similar to rough distributions in being wider and having lower peak counts than smooth distributions.

## 5.3 Comparison of Ice Roughness Estimates

The statistical parameters of each linear regression are presented in Table 6, along with the RMSE found between the observed and modelled values, where applicable.

### 395 5.3.1 Nez. - n

In Figure 10 the Manning's n determined by using the various representative roughness height metrics were compared to the Manning's n calculated using the Nezhikhovskiy relationship. These plots show that ice that was predicted to have a rougher sub-surface via the Nezhikhovskiy relationship also had a rougher surface, as observed using RPA-photogrammetry. The maximum peak height metric had, by far, the best fit based on the RMSE between the observed and modelled values.

However, the authors speculate that the maximum peak metric would be prone to influence from outliers. With this in mind, the 84th percentile of peaks had the next lowest RMSE between observed and modelled values. Linear regressions were computed for all metrics and are illustrated in their corresponding graphs. The parameters of these regressions are presented in Table 6. The RMSE between the data and the linear regression showed a less clear distinction, with all models performing well, although the SD and IQR had the lowest RMSE values. Considering the $R^2$ value of the linear regressions, the IQR had

the highest value. The $p$ statistic found that all relationships were significant ($\alpha \leq 0.05$) in this comparison. These results indicate the the ice roughness values derived from RPA-photogrammetry closely match roughness values predicted using the Nezhikhovskiy relationship.





**Table 5.** Categorization of ice roughness via cluster analysis and associated summary statistics

| Site | Date | Clus. | General Data | | Med. (m) | Peak Data | | 84th Perc. (m) |
|------|------|-------|--------------|--|----------|-----------|--|----------------|
| | | | IQR (m) | Kurt. (-) | | IQR (m) | Kurt. (-) | |
| DRLL06 | 11/21/2018 | 1 | 0.12 | 3.91 | 0.23 | 0.08 | 5.67 | 0.31 |
| DRLL06 | 2/20/2019 | 1 | 0.12 | 5.81 | 0.22 | 0.07 | 10.62 | 0.31 |
| DRLL06 | 11/12/2019 | 1 | 0.08 | 4.32 | 0.15 | 0.05 | 11.24 | 0.20 |
| DRLL06 | 11/13/2019 | 1 | 0.08 | 3.68 | 0.16 | 0.06 | 5.71 | 0.22 |
| DRLL06 | 11/26/2019 | 1 | 0.07 | 3.78 | 0.14 | 0.05 | 9.38 | 0.18 |
| DRLL03b | 11/14/2019 | 2 | 0.01 | 15.51 | 0.03 | 0.02 | 12.86 | 0.06 |
| DRLL05 | 11/12/2019 | 2 | 0.02 | 6.45 | 0.04 | 0.02 | 9.64 | 0.05 |
| DRLL05 | 11/13/2019 | 2 | 0.02 | 7.59 | 0.04 | 0.02 | 10.29 | 0.06 |
| DRLL08 | 11/13/2019 | 2 | 0.05 | 5.70 | 0.09 | 0.03 | 14.54 | 0.12 |
| DRLL08a | 11/26/2019 | 2 | 0.04 | 10.67 | 0.08 | 0.03 | 29.54 | 0.11 |
| Max. | | 1 | 0.12 | 5.81 | 0.23 | 0.08 | 11.24 | 0.31 |
| | | 2 | 0.05 | 15.51 | 0.09 | 0.03 | 29.54 | 0.12 |
| Mean | | 1 | 0.09 | 4.30 | 0.18 | 0.06 | 8.52 | 0.24 |
| | | 2 | 0.03 | 9.19 | 0.05 | 0.03 | 15.37 | 0.08 |
| Min. | | 1 | 0.07 | 3.68 | 0.14 | 0.05 | 5.67 | 0.18 |
| | | 2 | 0.01 | 5.70 | 0.03 | 0.02 | 9.64 | 0.05 |

### 5.3.2   Thickness - n

Since the only input parameter for the Nezhikhovskiy relationship is ice thickness, Figure 10 suggests relationship between
ice thickness and surface roughness. Since the original observations that supporting Equation 3 related thicker ice to ice with a
rougher sub-surface, the link between surface ice roughness and ice thickness supports a link between surface and sub-surface
ice roughness. Pursuant to the idea of a linkage between surface ice roughness and thickness, the observed Manning's ice
roughness was compared to the observed ice thickness in Figure 11. Linear regressions were computed for all metrics and are
illustrated in their corresponding graphs. The parameters of these regressions are presented in Table 6. All metrics showed a
strong positive relationship with ice thickness. The IQR again performed the best with the highest $R^2$ value and lowest RMSE,
although all metrics performed exceptionally well. The $p$ statistic shows that all relationships were significant ($\alpha \leq 0.05$) in
this comparison. Further research would be required to determine if this relationship may be used to estimate ice thickness
based on observed surface roughness.



**Table 6.** Performance statistics of applied linear models *\*RMSE between observed surface ice roughness and modelled subsurface ice roughness, errors are in the units of Manning's n*

| Model | Metric | Slope | $R^2$ | $F_{1,4}$ | p | RMSE | RMSE* |
|---|---|---|---|---|---|---|---|
| **Nez. - n** | SD | 0.492 | 0.930 | 53.217 | 0.002 | 0.001 | 0.038 |
| | IQR | 0.674 | 0.933 | 55.283 | 0.002 | 0.002 | 0.037 |
| | Min. Peak | 0.916 | 0.853 | 23.295 | 0.008 | 0.004 | 0.033 |
| | Max. Peak | 1.343 | 0.910 | 40.431 | 0.003 | 0.004 | 0.006 |
| | Mean Peak | 1.148 | 0.849 | 22.466 | 0.009 | 0.005 | 0.029 |
| | 84th of Peaks | 1.339 | 0.887 | 31.439 | 0.005 | 0.004 | |
| **Thick. - n** | SD | 0.014 | 0.986 | 284.096 | <0.000 | 0.001 | 0.025 |
| | IQR | 0.019 | 0.992 | 496.959 | <0.001 | 0.001 | - |
| | Min. Peak | 0.026 | 0.953 | 81.092 | 0.001 | 0.002 | - |
| | Max. Peak | 0.037 | 0.970 | 128.176 | <0.001 | 0.002 | - |
| | Mean Peak | 0.033 | 0.954 | 82.852 | 0.001 | 0.002 | - |
| | 84th of Peaks | 0.038 | 0.971 | 135.185 | <0.001 | 0.002 | - |

## 5.4 Alternative Uses for RPA-Photogrammetry

A common problem in river ice elevation surveys is the selection of a single, representative value to define the average ice surface elevation in a given area. It is up to the observer to use their best judgement to visually select a single representative point to survey while walking along the river bank. During massive ice jam events, this field task is dangerous, and frequently impossible. The above research shows that RPA-photogrammetry can be used to accurately survey ice areas for the purpose of observing local topography, with much lower risks to field personnel than traditional ice surveying methods. Once a DEM

has been established, a representative ice elevation value can be determined using linear modelling. From an appropriately selected sub-area, a linearly-defined plane-of-best-fit may be determined, the mean value of which would provide an excellent estimation of a locally representative ice elevation. This analysis could be extended to examine shear walls, and determine maximum ice elevation of a freeze-up jam, after the fact.

## 6 Conclusions

The research presented in this paper has developed novel methods for the capture of fluvial ice surface roughness and the analysis of the resultant high-resolution data. Through field trials and controlled land-based experiments, it was determined that RPA-photogrammetry produced an accurate digital representation of rough or smooth ice covers, with a maximum vertical error of 0.06 m if using three or more ground control points over a 200 m by 100 m area. For the sole purpose of roughness characterization, it was determined that geo-rectification were unnecessary using our equipment. The relatively inexpensive



consumer-grade RPA was able to operate in harsh winter field conditions, with an approximately 50% reduced battery life. Using this novel, high-resolution data, bulk statistical properties including SD, skewness, kurtosis, and IQR were calculated for two classifications of ice covers.

The hypothesis of this research, that the surface roughness of a newly-frozen fluvial ice cover is indicative of subsurface roughness was tested. When comparing the results of the Nezhikhovskiy relationship to the Manning's n values determined through RPA-photogrammetry, all roughness height metrics produced significant relationships, many with $R^2 > 0.9$. In this comparison the IQR had the lowest $p$ value and highest $R^2$ of 0.933 and 0.002 respectively. Aside from the maximum peak value, the 84$^{th}$ percentile of peaks values had the lowest RMSE when compared to the one-to-one line.

Positive correlation was observed between surface roughness and ice thickness across all tested metrics. Once again the IQR performed the best with the highest $R^2$ and lowest RMSE and $p$ values (0.992, $< 0.001$, and 0.001 respectively). This relationship provides a basis for a link between surface and sub-surface ice roughness, through the Nezhikhovskiy relationship.

Through evaluation of the statistical properties of the distribution of DEM heights observed via RPA-photogrammetry, several interesting patterns were found. All distributions were found to be non-normal, when evaluated with the Shapiro-Wilk normality test, however they display a qualitatively normal appearance. The IQR for these cases were often very similar. k-means cluster analysis displayed two strong groupings, one comprised of DRLL06, and the other of DRLL03b, DRLL05, DRLL08, and DRLL08a. Based on field observations the first group was labelled "rough" ice, while the second was labelled "smooth" ice. The range of IQR values observed in each group was 0.01 - 0.05 m and 0.07 - 0.12 m for the smooth and rough groups respectively.

The methods presented in this research can conceivably be applied to further uses in the field of fluvial ice monitoring. The high-resolution DEMs produced by this method, and retrodictive nature of the data provide ample opportunity for the development of novel analysis methods, and the replacement of traditional techniques. Ice profile surveys may be aided by using RPA-photogrammetry to determine a more representative plane-of-best-fit for a spot measurement of ice elevation. Shear walls may be captured and analysed in their entirety, even immediately after, or during break-up. This research also presents a possible link between surface ice roughness and ice thickness, which may provide for a method of estimating ice thickness using RPA-photogrammetry. The use of RPA-photogrammetry for the monitoring of fluvial ice covers offers a quicker, safer, and cheaper alternative to any previous method of high-resolution topographic data collection, and it's applications are open for development.

*Author contributions.* JE, SC, and AW developed the research plan and conducted field work. JE processed all field data and developed all data analysis tools. JE and SC drew conclusions based on results.

*Competing interests.* The authors declare that they have no conflict of interest.



*Data availability.* Data related to this study including raw images, point clouds, and summary data are available upon request from the corresponding author.

*Acknowledgements.* Project funding provided by NSERC (IRCPJ472185-13) and Manitoba Hydro (G274/P274) as well as an NSERC CGS-M scholarship was instrumental in facilitating this research.



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





**Figure 8.** Examples of three types of roughness observed in ice surface roughness samples along the indicated transects (dotted lines), DRLL08a 2019.11.13



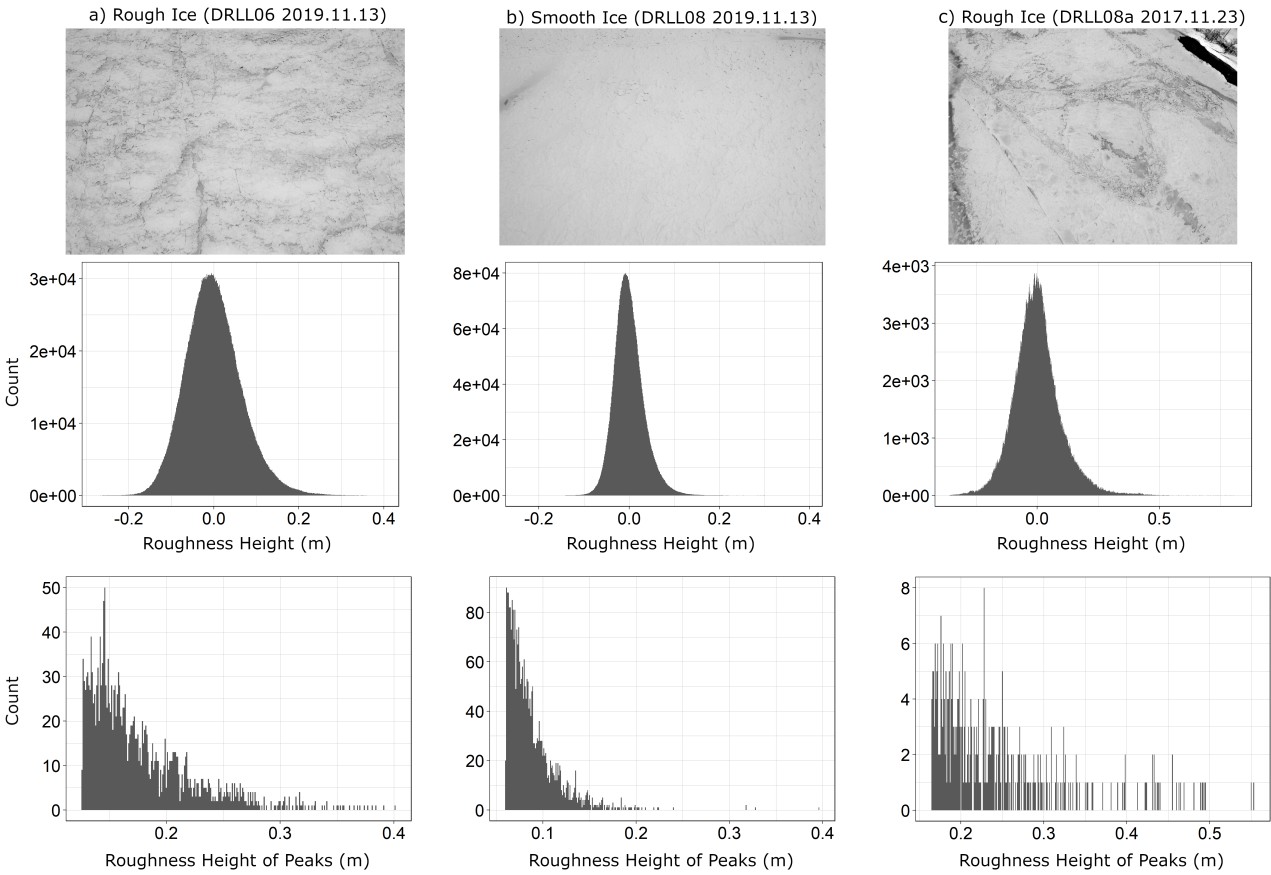

**Figure 9.** Comparison of representative roughness height histograms across three identified ice types







Figure 10. Comparison of observed Manning's ice roughness calculated from various metrics to calculated Manning's ice roughness using the Nezhikhovskiy relationship





**Figure 11.** Comparison of observed Manning's ice roughness calculated from various metrics to observed ice thickness