# Peer review of "Ice Roughness Estimation via Remotely Piloted Aircraft and Photogrammetry"

_The Cryosphere, 2021_

## Author Response (AR1)

| Reviewer 1 Comments | Formal response to reviewer |
|---|---|
| Presentation of the manuscript, especially the methodology and discussion is not clear. Hypothesis validation logic is hard to follow. | Addressed through revisions and rewritting of the main text |
| Writing can be improved significantly. | Addresssed through revisions and rewritting of the main text |
| The section 2 seems a mixture of site description, metrics of surface roughness, methods used for estimating surface roughness, and the RPA. I suggest separating the part of site description to be a new section and moving the rest to the methodology, to make the manuscript's structure clearer | Subsections added to section 2 |
| In section 4.1, the "two points" scenario has the worst performance, compared with the other scenarios using more control points. Does it indicate that at least three control points should be used to geo-rectify the DEM results? If it does, why the results of using no control points are comparable to the geo-rectified ones, as presented in Table 2? Why their difference can indicate the systematic errors introduced by the lack of geo rectification? | Additional text added to section to clarify. In essence, the data gathered using the RPA in the land-based test scenarios required geo-rectification to make a meaningful comparison to the data collected with the GNSS survey equipment. For the comparison of data collected at the river sites, the exact elevation of the ice cover was not of concern, rather the texture and roughness height. The elevation of the ice cover, and any large trends in it were removed during the filtering stage. |
| In section 3.2.4, you compare the Nezhikhovskiy ice Manning's n and the observed ice Manning's n. what is the observed ice Manning's n? Is it another observable you estimate from the RPA images? Have you ever checked the Manning's n estimated by Beltaos equation? | Additional text was added to the section to clarify. The observed ice Manning's n is the value derived from the RPA data using the methods outlined in section 3.2.3 (observed DEM -> filtered DEM -> statistical values representing roughness height (SD, IQR, etc...) -> Manning's n calculated from the statistical values using the Beltaos equation. |
| In Line 275, you demonstrate that the comparison between the Nezhikhovskiy ice Manning's n and the observed ice Manning's n can indicate the relationship between surface and subsurface roughness. Accordingly, in section 5.3.1, you present thecomparison results. Could you elaborate on that why this comparison can indicate the linkage between surface and subsurface roughness? | Additional text was added to the section to clarify. Since the Nezhikovskiy equation illustrates a relationship with ice thickness and subsurface ice roughness, a relationship between surface ice roughness (as observed by the RPA) and ice thickness would imply a further relationship between surface and subsurface ice roughness. |
| In Figure 10, the label of the horizontal axis is PRA roughness. Is this consistent with the observed ice Manning's n? | The figure axis was clarified to "RPA Observed Ice Surface Manning's n", and the y-axis has been renamed to "Nezhikovskiy Predicted Subsurface Ice Manning's n". |
| Could you elaborate on this sentence "Since the original observations that supporting Equation 3 related thicker ice to ice with a rougher sub-surface, the link between surface ice roughness and ice thickness supports a link between surface and sub-surface ice roughness." in the section of 5.3.2? | The text has been expanded in Section 5.3.2, as well as previous sections where this possible relationship has been brought up. |

| | |
|---|---|
| The Beltaos equation requires the roughness height of the flow boundary (denoted by D) and hydraulic radius (R). The roughness height of the flow boundary refers to the roughness of the upper ice surface or the lower one? | Changes to the text have been made to clarify this. The original formulation of the Beltaos equation requires the roughness height (D) of the subsurface of an ice cover. Since the hypothesis is that the roughness of the upper ice surface (surface) and the lower ice surface (subsurface) are the same, the roughness height observed with the RPA is used in the Beltaos equation to determine the observed ice Manning's n, which is then compared to the ice Manning's n predicted by the Nezhikhovskiy equation. |
| In section 3.1.2, how to use the observed water level and channel bathymetry data to estimate hydraulic radius? Similarly, what kind of observation and method used to estimate ice thickness? The references are at least included. | The text has been expanded to clarify. The hydraulic radius was estimated using a 1D at-a-station hydraulic model based on Manning's equation. Ice thickness was determined through a combination of measurements using ice-coring equipment when safe, and visual observations from field visits, photos, and ice-transects. |
| Line 255, what is the peak picking algorithm? | Text has been clarified, the algorithm used is "peakpick" available in R through the peakPick package. |
| What does "i" mean in Equation 2? | "i" refers to ice, to distinguish the Manning's n associated with ice from the usual application of Manning's n, which is stream bed roughness. Clarification has been added to the text. |
| Line 275, what does "p" stand for? | In the context of this research, and through the evaluation of the significance of the applied linear models, the p value is a measure of the probability that the data show a significant trend through random chance, and not through an actual relationship. The text has been expanded to clarify this. |
| Table 6, What is F1,4? | Text has been expanded to clarify. The F statistic is another commonly reported criteria for the evaluation of linear regressions. It should be compared to the Critical F value (now included in the text). The F statistic also usually reports the degrees of freedom of it's components, which in this study is 1 and 4. |

**Reviewer 2 Comments**

The authors state in the abstract that the central hypothesis of the paper is that "the surface roughness of a newly-frozen fluvial ice cover is indicative of the subsurface roughness." However, in the body of the manuscript, they jump immediately into the details of how surface roughness is calculated, almost without discussing the hypothesis at all. Somewhere in the introduction or the background sections, the authors need to discuss their rationale for this hypothesis. What is the physical reason for suspecting that surface and subsurface roughness are connected? As an outsider from this field, it seems to me that the surface and subsurface of the ice cover are subject to very different environmental conditions. The authors have data from this case study that seems to indicate that the surface and subsurface roughness are linked, but they should explain why they expect this to be the case in general. Also, the authors should indicate from earlier on (possibly in the abstract) that "subsurface roughness" refers to the water-ice interface. At first I wasn't sure if that was the case, or if they were referring to the roughness of the underlying riverbed.

Significant changes have been made to the wording and composition of the abstract and introduction to clarify and eloborate the introduction and hypothesis of the research.

The estimates of surface roughness from the drone seem to correlate well with estimates of subsurface roughness from the Nezhikhovskiy equation. This seems to bethe most important outcome of the paper, and it shows that the method is promising, but it is also based on just a handful of sites along a single river. In the Introduction and Discussion sections, I think the authors should state that this is a preliminary test of the Method, and additional data from other field sites is necessary in the future to determine how effective it truly is.

Text has been added to indicate that these results reprsent a first attempt, and require further validation.

In its present form, the writing in the paper is often awkward and unclear, which takes away somewhat from the message. Grammatical errors are common (for example, "its" and "it's" are often confused), the language often seems unnecessarily wordy, and some sentences are even left incomplete. I think the authors should do a close reading of the manuscript to improve the clarity of the writing.

The manuscript has been thouroughly proof-read to improve the grammar and clarify the message.

Specific Comments:

| | |
|---|---|
| The abstract seems to be longer than necessary. For example, the first three sentences provide background information about photogrammetry which is more fitting in the Introduction or Background sections. The authors should try to get to the point more quickly, which is that they are testing a method using drones to estimate the ice-water roughness, and the method is valuable because it is less hazardous than the more traditional route of measuring ice thickness in order to apply the Nezhikhovskiy equation. | The abstract has been shortenend. |
| The Background section is difficult to follow as it lacks a clear structure, lumping together a description of the field site, details about traditional methods for estimating roughness and the Mannings coefficient, and details about photogrammetry. I think this section should be broken into at least three subsections with appropriate titles. | Subsections have been added, and the information has been somewhat distilled. |
| Since the authors mention that several methods exist for constructing DEMs from overlapping images in lines 129-130, please briefly describe how Structure from Motion differs from the other methods. | The author was mistaken, while some other methods exist they are very niche in their application and not relevant to the paper. The text has been changed to reflect this. |
| Please briefly define doming errors at the start of the paragraph that begins in line 132. | Doming errors are caused by lens distortion in the camera mounted to the RPA. These errors are addressed through an automatic calibration process within AgiSoft (the program used for the evaluation of photogrammetry in this study). The authors found that the discussion of doming errors within this text was outside of the scope of the paper, and it was removed. |
| What is the rationale behind the codes for the study locations (e.g., DRLL08)? For the purposes of this paper, would it be appropriate to give them simpler names, like Site 1, Site 2, etc? | Text added to section 2 to clarify site naming, superfluous sites removed from keymap. Dauphin River Levelogger (DRLL) prefix is kept throughout the paper to allow for easy cross-referencing between other papers regarding this study site. |

Please add more explanation to the paragraph that runs from lines 236-247. I understand that you are removing certain trends from the DEM to better quantify roughness, but I find it hard to follow the details. Please provide more details about the plots in Figs 7a and 7b, and how these plots were used to assign the low-pass and high pass components of the filter

The figure was simplified to streamline the explanation in the text. The text has also been clarified. The components of the filter were selected through an extentsive iterative process of visual analysis of the image. Wavelength values which were below or above the ideal values caused obvious edge-cutoff errors, or insufficient trend removal.

The beginning of the paragraph that starts at line 270 is critical but unclear. What is "observed ice Manning's n"? Is this the Mannings coefficient as calculated using equation (1) or (3)? Please be very clear here. The first few sentences of this paragraph are difficult to follow and the sentence beginning on line 271 is incomplete.

This paragraph has been reworked and expanded on to make the text more clear and correct.